# Global prediction of extreme floods in ungauged watersheds

Grey Nearing[1✉], Deborah Cohen[1], Vusumuzi Dube[1], Martin Gauch[1], Oren Gilon[1], Shaun Harrigan[2], Avinatan Hassidim[1], Daniel Klotz[3], Frederik Kratzert[1], Asher Metzger[1], Sella Nevo[4], Florian Pappenberger[2], Christel Prudhomme[2], Guy Shalev[1], Shlomo Shenzis[1], Tadele Yednkachw Tekalign[1], Dana Weitzner[1] & Yossi Matias[1]

Floods are one of the most common natural disasters, with a disproportionate impact in developing countries that often lack dense streamflow gauge networks[1]. Accurate and timely warnings are critical for mitigating flood risks[2], but hydrological simulation models typically must be calibrated to long data records in each watershed. Here we show that artificial intelligence-based forecasting achieves reliability in predicting extreme riverine events in ungauged watersheds at up to a five-day lead time that is similar to or better than the reliability of nowcasts (zero-day lead time) from a current state-of-the-art global modelling system (the Copernicus Emergency Management Service Global Flood Awareness System). In addition, we achieve accuracies over five-year return period events that are similar to or better than current accuracies over one-year return period events. This means that artificial intelligence can provide flood warnings earlier and over larger and more impactful events in ungauged basins. The model developed here was incorporated into an operational early warning system that produces publicly available (free and open) forecasts in real time in over 80 countries. This work highlights a need for increasing the availability of hydrological data to continue to improve global access to reliable flood warnings.

Floods are the most common type of natural disaster[3] and the rate of flood-related disasters has more than doubled since 2000[4]. This increase in flood-related disasters is driven by an accelerating hydrological cycle caused by anthropogenic climate change[5,6]. Early warning systems are an effective way to mitigate flood risks, reducing flood-related fatalities by up to 43%[7,8] and economic costs by 35–50%[9,10]. Populations in low- and middle-income countries make up almost 90% of the 1.8 billion people that are vulnerable to flood risks[1]. The World Bank has estimated that upgrading flood early warning systems in developing countries to the standards of developed countries would save an average of 23,000 lives per year[2].

In this paper, we evaluate the extent to which artificial intelligence (AI) trained on open, public datasets can be used to improve global access to forecasts of extreme events in global rivers. On the basis of the model and experiments described in this paper, we developed an operational system that produces short-term (7-day) flood forecasts in over 80 countries. These forecasts are available in real time without barriers to access such as monetary charge or website registration (https://g.co/floodhub).

A major challenge for riverine forecasting is that hydrological prediction models must be calibrated to individual watersheds using long data records[11,12]. Watersheds that lack stream gauges to supply data for calibration are called ungauged basins, and the problem of 'prediction in ungauged basins' (PUB) was the decadal problem of the International Association of Hydrological Sciences (IAHS) from 2003 to 2012[13]. At the

end of the PUB decade, the IAHS reported that little progress had been made against the problem, stating that "much of the success so far has been in gauged rather than in ungauged basins, which has negative effects in particular for developing countries"[14].

Only a few per cent of the world's watersheds are gauged, and stream gauges are not distributed uniformly across the world. There is a strong correlation between national gross domestic product and the total publicly available streamflow observation data record in a given country (Extended Data Fig. 1 shows this log–log correlation), which means that high-quality forecasts are especially challenging in areas that are most vulnerable to the human impacts of flooding.

In previous work[15], we showed that machine learning can be used to develop hydrological simulation models that are transferable to ungauged basins. Here we develop that into a global-scale forecasting system with the goal of understanding scalability and reliability. In this paper, we address whether, given the publicly available global streamflow data record, it is possible to provide accurate river forecasts across large scales, especially of extreme events, and how this compares with the current state of the art.

The current state of the art for real-time, global-scale hydrological prediction is the Global Flood Awareness System (GloFAS)[16,17]. GloFAS is the global flood forecasting system of Copernicus Emergency Management Service (CEMS), delivered under the responsibility of the European Commission's Joint Research Centre and operated by the European Centre for Medium-Range Weather Forecasts (ECMWF) in

[1]Google, https://research.google/. [2]European Centre for Medium-Range Weather Forecasts, Reading, UK. [3]Helmholtz Centre for Environmental Research - UFZ, Leipzig, Germany. [4]RAND Corporation, Los Angeles, CA, USA. ✉e-mail: nearing@google.com

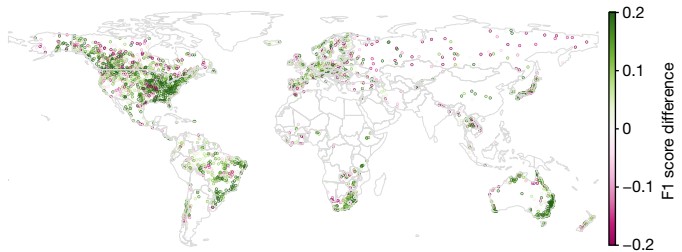

**Fig. 1 | Differences between nowcast (0-day lead time) F1 scores for 2-year return period events between our AI model and GloFAS over the period 1984–2021.** The AI model improves over GloFAS in 70% of gauges (*N* = 3,673). GloFAS simulation data from the Climate Data Store[33]. Basemap from GeoPandas[34].

its role of CEMS Hydrological Forecast Centre – Computation. We use GloFAS version 4, which is the current operational version that went live in July 2023. Other forecasting systems exist for different parts of the world[18–20], and many countries have national agencies responsible for producing early warnings. Given the severity of impacts that floods have on communities around the world, we consider it critical that forecasting agencies evaluate and benchmark their predictions, warnings and approaches, and an important first step towards this goal is archiving historical forecasts.

## AI improves forecast reliability

The AI model developed for this study uses long short-term memory (LSTM) networks[21] to predict daily streamflow through a 7-day forecast horizon. The model is described in detail in Methods, and a version of the model suitable for research is implemented in the open-source NeuralHydrology repository[22]. Input, target and evaluation data are described in Methods.

This AI forecast model was trained and tested out-of-sample using random *k*-fold cross-validation across 5,680 streamflow gauges. Other types of cross-validation experiment are reported in Methods (that is, by withholding all gauges in terminal watersheds, entire climate zones or entire continents). In addition, all metrics reported for the AI model were calculated with streamflow gauge data from time periods not present in training (in addition to stream gauges that were not present in training), meaning that cross-validation splits were out-of-sample across time and location. By contrast, metrics for GloFAS were calculated over a combination of gauged and ungauged locations, and over a combination of calibration and validation time periods. This means that the comparison favours the GloFAS benchmark. This is necessary because calibrating GloFAS is computationally expensive to the extent that it is not feasible to re-calibrate over cross-validation splits.

Our objective is to understand the reliability of forecasts of extreme events, so we report precision, recall and F1 scores (F1 scores are the harmonic mean of precision and recall) over different return period events. Other standard hydrological metrics are reported in Methods. Statistical tests are described in Methods.

Figure 1 shows the global distribution of F1 score differences for 2-year return period events at a 0-day lead time over the period 1984–2021 (*N* = 3,360). Lead time is expressed as the number of days from the time of prediction, such that a 0-day lead time means that streamflow predictions are for the current day (nowcasts). The AI model improved over (was at least equivalent to) GloFAS version 4 in 64% (65%), 70% (73%), 60% (73%) and 49% (76%) of gauges for return period events of 1 year (*N* = 3,638, *P* = 6 × 10⁻⁸⁷, Cohen's *d* = 0.22), 2 years (*N* = 3,673, *P* < 3 × 10⁻¹⁸¹, *d* = 0.41), 5 years (*N* = 3,360, *P* = 8 × 10⁻¹³⁰, *d* = 0.42) and 10 years (*N* = 2,920, *P* < 1 × 10⁻⁶⁶, *d* = 0.33).

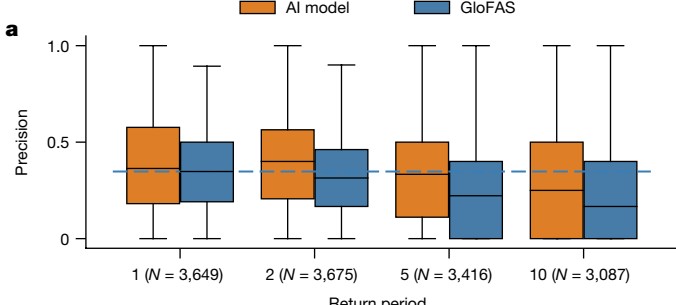

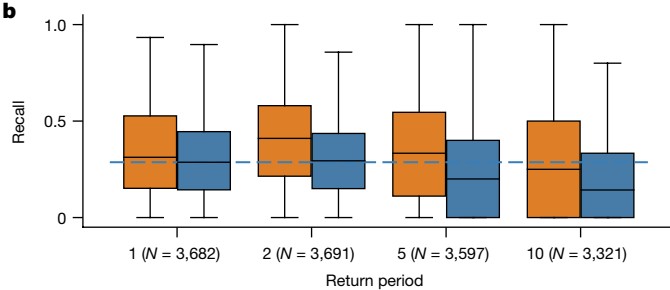

**Fig. 2 | Distributions over nowcast (0-day lead time) precision and recall as a function of return period. a,b,** The AI model is more reliable, on average, over all return periods. The AI model has precision over 5-year return period events that is not statistically different to GloFAS over 1-year return period events, and recall that is better than GloFAS over 1-year return period events. Statistical tests are reported in the main text. The boxes show distribution quartiles and whiskers show the full range excluding outliers. The blue dashed line is the median score for GloFAS over 1-year events and is plotted as a reference. Tick labels indicate the sample size (number of gauges) for each boxplot; precision scores (**a**) and recall scores (**b**) were calculated over slightly different gauge groups in cases where there are no events of a given magnitude at a given gauge location in either the observations or model predictions causing one score for one model to be undefined. GloFAS and the AI model are always compared over an identical set of gauges in all cases. GloFAS simulation data from the Climate Data Store[33].

## Return periods

More extreme hydrological events (that is, events with larger return periods) are both more important and (when using classical hydrology models) typically more difficult to predict. A common concern[23–26] about using AI or other types of data-driven approach is that reliability might degrade over events that are rare in the training data. There is prior evidence that this concern might not be valid for streamflow modelling[27].

Figure 2 shows the distributions over precision and recall for different return period events. The AI model has higher precision and recall scores for all return periods (*N* > 3,000, *P* < 1 × 10⁻⁵), with effect sizes ranging from *d* = 0.15 (1-year precision scores) to *d* = 0.46 (2-year recall scores). Differences between precision scores from the AI model over 5-year return period events and from GloFAS over 1-year return period events are not significant at *α* = 1% (*N* = 3,465, *P* = 0.02, *d* = −0.01), and recall scores from the AI model for 5-year events are better than GloFAS recall scores for 1-year events (*N* = 3,586, *P* = 1 × 10⁻¹⁸, *d* = 0.20).

## Forecast lead time

Figure 3 shows F1 scores over lead times through the 7-day forecast horizon for return periods between 1 year and 10 years. Compared with GloFAS nowcasts (0-day lead time), AI forecasts have either better or not statistically different reliability (F1 scores) up to a 5-day lead time for 1-year (AI is significantly better; *N* = 2,415, *P* = 6 × 10⁻⁶, *d* = 0.08), 2-year (no statistical difference; *N* = 2,162, *P* = 0.98, *d* = 2 × 10⁻⁴) and 5-year (no statistical difference; *N* = 1,298, *P* = 0.69, *d* = 0.025) return period events.

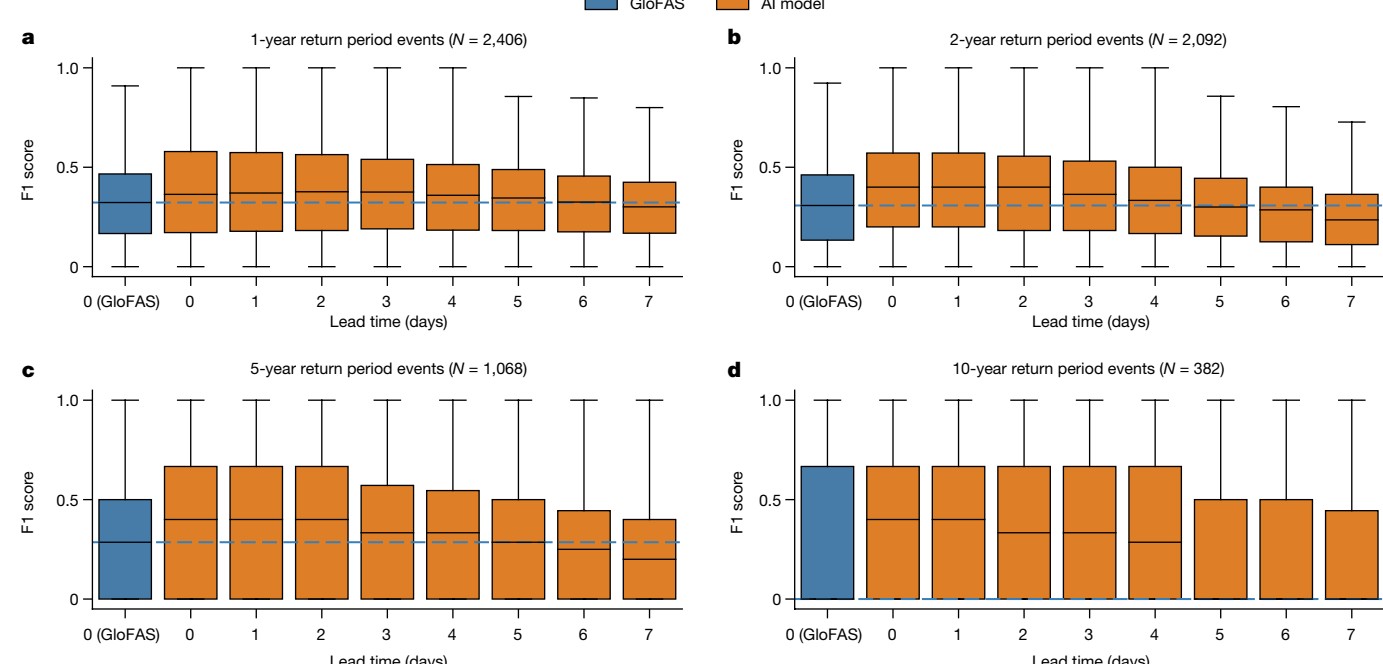

**Fig. 3 | Distributions over F1 scores at all evaluation gauges as a function of lead time for different return periods. a–d**, The AI model has F1 scores over 1-year (**a**), 2-year (**b**), 5-year (**c**) and 10-year (**d**) return period events at up to 5-day lead times that are either statistically better than or not statistically different to GloFAS over the same events at 0-day lead time. Statistical tests are reported in the main text. The boxes show distribution quartiles and whiskers show the full range excluding outliers. The blue dashed line is the median score for GloFAS nowcasts and is plotted as a reference. GloFAS simulation data from the Climate Data Store[33].

### Continents

Both models show differences in reliability in different areas of the world. Over 5-year return period events, GloFAS has a 54% difference between mean F1 scores in the lowest-scoring continent (South America, F1 = 0.15) and the highest-scoring continent (Europe, F1 = 0.32), meaning that, on average, true positive predictions are twice as likely (at a proportional rate). The AI model also has a 54% difference between mean F1 scores in the lowest-scoring continent (South America, F1 = 0.21) and the highest-scoring continent (Southwest Pacific: F1 = 0.46), which is due mostly to a large increase in skill in the Southwest Pacific relative to GloFAS ($d = 0.68$).

Figure 4 shows the distributions of F1 scores over continents and return periods. The AI model has higher scores in all continents and return periods ($P < 1 \times 10^{-2}$, $0.10 < d < 0.68$) with three exceptions where there is no statistical difference: Africa over 1-year return period events ($P = 0.07$, $d = 0.03$) and Asia over 5-year ($P = 0.04$, $d = 0.12$) and 10-year ($P = 0.18$, $d = 0.12$) return period events.

### Predictability of forecast reliability

A challenge to forecasting in ungauged basins is that there is often no way to evaluate reliability in locations without ground-truth data. A desirable quality of a model is that forecast skill should be predictable from other observable variables, such as mapped or remotely sensed geographical and/or geophysical data. In addition, although AI-based forecasting offers better reliability in most places, this is not the case everywhere. It would be beneficial to be able to predict where different models can be expected to be more or less reliable.

We have found that it is difficult to use catchment attributes (geographical, geophysical data) to predict where one model performs better than another. Extended Data Fig. 2 shows a confusion matrix from a random forest classifier trained on a subset of HydroATLAS attributes[28] that predicts whether the AI model or GloFAS performs better (or similar) in each individual watershed. The classifier was trained with stratified $k$-fold cross-validation and balanced sampling, and usually predicts that the AI model is better (including in 70% of cases where GloFAS is actually better). This indicates that it is difficult to find systematic patterns about where each model is preferable, based on available catchment attributes.

However, it is possible to predict, with some skill, where an individual model will perform well versus poorly. As an example, Fig. 5 shows confusion matrices from random forest classifiers that predict whether F1 scores for out-of-sample gauges (effectively ungauged locations) will be above or below the mean over all evaluation gauges. Both models (the AI model and GloFAS) have similar overall predictability (71% micro-averaged precision and recall for GloFAS and 73% for the AI model).

Feature importances from these reliability classifiers are shown in Extended Data Fig. 3. Feature importance is an indicator about which geophysical attributes determine high versus low reliability (that is, what kind of watersheds do these models simulate well versus poorly). The most important features for the AI model are: drainage area, mean annual potential evapotranspiration (PET), mean annual actual evapotranspiration (AET) and elevation, whereas the most important features for GloFAS were PET and AET. Correlations between attributes and reliability scores are generally low, indicating a high degree of nonlinearity and/or parameter interaction.

AET and PET are (inverse) indicators of aridity, and hydrology models usually perform better in humid basins because peaky hydrographs that occur in arid watersheds are difficult to simulate. This effect is present for both models. The AI model is more correlated with basin size (drainage area) and generally performs better in smaller basins. This indicates a way that machine-learning-based streamflow modelling might be improved, for example, by focusing training or fine-tuning on larger basins, or by implementing an explicit routing or graph model to allow for direct modelling of subwatersheds or smaller hydrological response units—for example, as outlined in ref. 29.

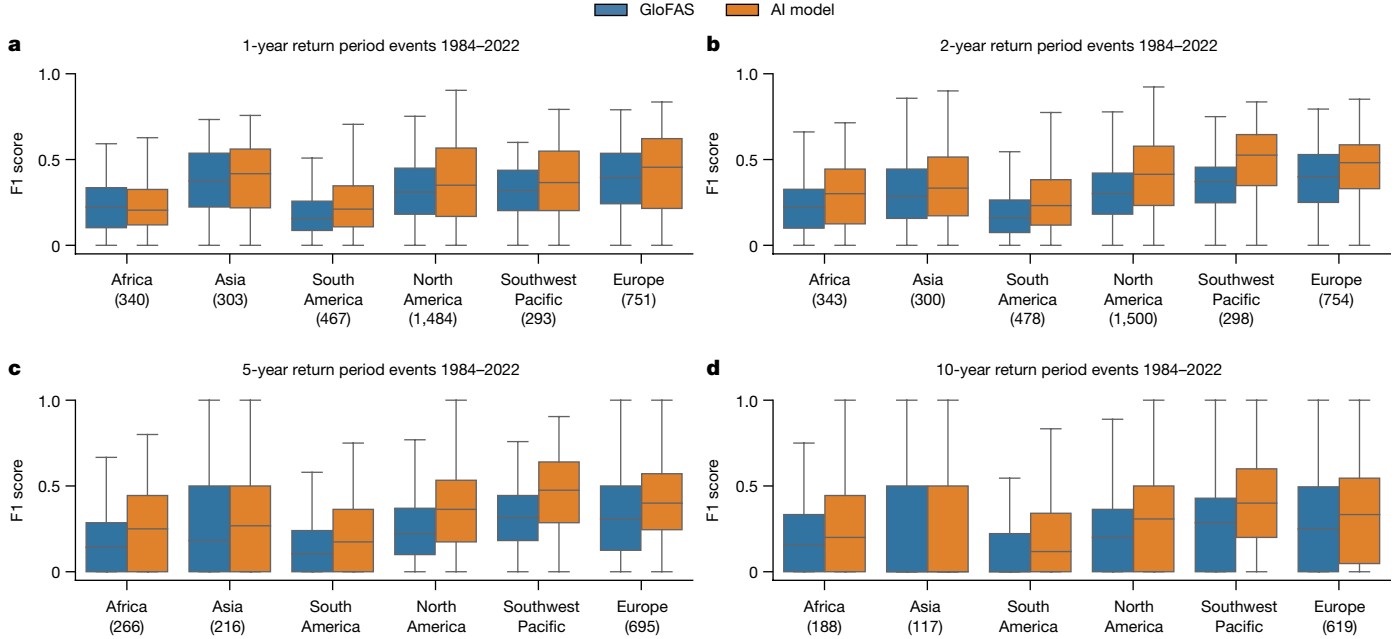

**Fig. 4 | F1 score distributions over different continents and return periods.** **a**–**d**, The AI model has higher scores in all continents over 1-year (**a**), 2-year (**b**), 5-year (**c**) and 10-year (**d**) return period events with three exceptions where there is no statistical difference: Africa over 1-year return period events and Asia over 5-year and 10-year return period events. Both models have large location-based differences between reliability that could be addressed by increasing global access to open hydrological data. Statistical tests are reported in the main text. The boxes show distribution quartiles and whiskers show the full range excluding outliers. GloFAS simulation data from the Climate Data Store[33].

A global map of the predicted skill from a regression (rather than classifier) version of this random forest skill predictor is shown in Fig. 6 for 1.03 million level-12 HydroBASINS watersheds[30]. This gives some indication about where a global version of the ungauged AI forecast model is expected to perform well.

## Conclusion and discussion

Although hydrological modelling is a relatively mature area of study, areas of the world that are most vulnerable to flood risks often lack reliable forecasts and early warning systems. Using AI and open datasets, we are able to significantly improve the expected precision, recall and lead time of short-term (0–7 days) forecasts of extreme riverine events. We extended, on average, the reliability of currently available global nowcasts (lead time 0) to a lead time of 5 days, and we were able to use AI-based forecasting to improve the skill of forecasts in Africa to be similar to what are currently available in Europe.

Apart from producing accurate forecasts, another aspect of the challenge of providing actionable flood warnings is dissemination of those warnings to individuals and organizations in a timely manner. We support the latter by releasing forecasts publicly in real time, without cost or barriers to access. We provide open-access real-time forecasts to support notifications—for example, through the Common Alerting Protocol and push alerts to personal smartphones, and through an open online portal at https://g.co/floodhub. All of the reanalysis and reforecasts used for this study are included in an open-source repository, and a research version of the machine-learning model used for this study is available as part of the open-source NeuralHydrology repository on GitHub[22].

There is still a lot of room to improve global flood predictions and early warning systems. Doing so is critical for the well-being of millions of people worldwide whose lives (and property) could benefit from timely, actionable flood warnings. We believe that the best way to improve flood forecasts from both data-driven and conceptual modelling approaches is to increase access to data. Hydrological data are

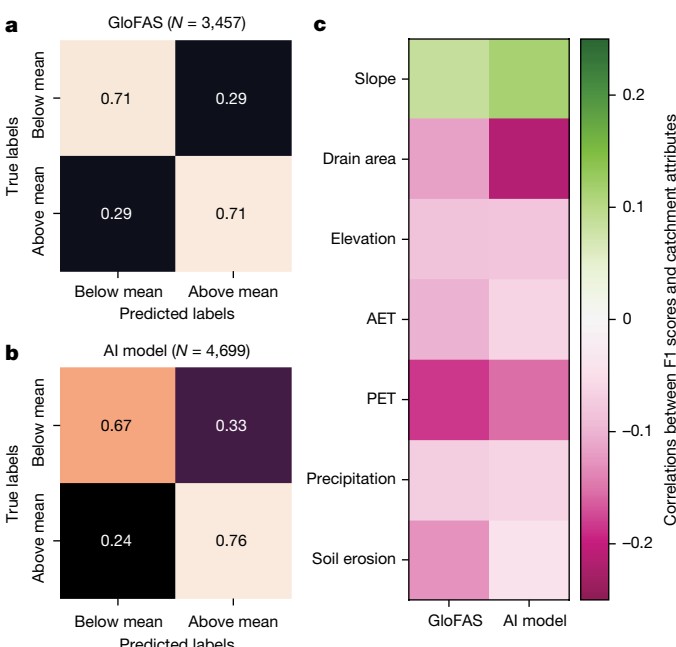

**Fig. 5 | Testing the ability to predict whether a given model will perform above or below average at any given location. a**,**b**, Confusion matrices of out-of-sample predictions about whether F1 scores from GloFAS (**a**) and the AI model (**b**) at each gauge are above or below the mean F1 score from the same model over all gauges. The numbers shown on the confusion matrices are micro-averaged precision and recall, and the colours serve as a visual indication of these same numbers. **c**, Correlations between F1 scores and HydroATLAS catchment attributes that have the highest feature importance ranks from these trained score classifier models. GloFAS simulation data from the Climate Data Store[33].

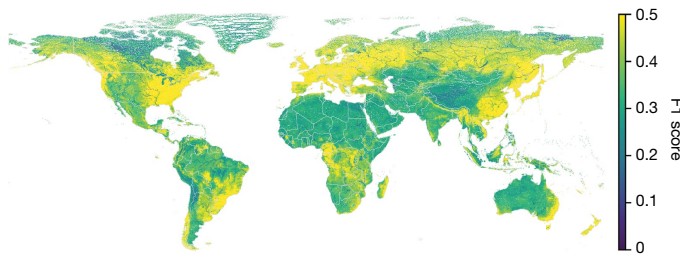

**Fig. 6 | Global predicted skill.** This map shows predictions of 2-year return period F1 scores over 1.03 million HydroBASINS level-12 watersheds for the AI forecast model. Basemap from GeoPandas[34].

required for training or calibrating accurate hydrology models, and for updating these models in real time (for example, through data assimilation[31]). We encourage researchers and organizations with access to streamflow data to contribute to the open-source Caravan project at https://github.com/kratzert/Caravan[32].

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

## Methods

### AI model

The AI streamflow forecasting model reported in this paper extends work in ref. 35, which developed hydrological nowcast models using LSTM networks that simulate sequences of streamflow data from sequences of meteorological input data. Building on that, we developed a forecast model that uses an encoder–decoder model with one LSTM running over a historical sequence of meteorological (and geophysical) input data (the encoder LSTM) and another, separate, LSTM that runs over the 7-day forecast horizon with inputs from meteorological forecasts (the decoder LSTM). The model architecture is illustrated in Extended Data Fig. 4.

The model uses a hindcast sequence length of 365 days, meaning that every forecast sequence (0–7 days) saw meteorological input data from the preceding 365 days and meteorological forecast data over the 0–7-day forecast horizon. We used a hidden size of 256 cell states for both the encoder and decoder LSTMs, a linear-cell-state transfer network and a nonlinear (fully connected layer with hyperbolic tangent activation functions) hidden-state transfer network. The model was trained on 50,000 minibatches with a batch size of 256. All inputs were standardized by subtracting the mean and dividing by the standard deviation of training-period data.

The model predicts, at each time step, (time-step dependent) parameters of a single asymmetric Laplacian distribution over area-normalized streamflow discharge, as described in ref. 36. The loss function is the joint negative log-likelihood of that heteroscedastic density function. To be clear, the model predicts a separate asymmetric Laplacian distribution at each time step and each forecast lead time. The results reported in this paper were calculated over a hydrograph that results from averaging the predicted hydrographs from an ensemble of three separately trained encoder–decoder LSTMs. The hydrograph from each of these separately trained LSTMs is taken as the median (50th percentile) flow value from the predicted Laplacian distribution at each time step and forecast lead time.

Using the dataset described herein, the AI model takes a few hours to train on a single NVIDIA-V100 graphics processing unit. The exact wall time depends on how often validation is done during training. We use 50 validation steps (every 1,000 batches), resulting in a 10-hour train time for the full global model.

### Input data

The full dataset includes model inputs and (streamflow) targets for a total of 152,259 years from 5,680 watersheds. The total size of the dataset saved to disk (including missing values in a dense array) is 60 GB.

Input data came from the following sources.
- Daily-aggregated single-level forecasts from the ECMWF Integrated Forecast System (IFS) High Resolution (HRES) atmospheric model. Variables include: total precipitation (TP), 2-m temperature (T2M), surface net solar radiation (SSR), surface net thermal radiation (STR), snowfall (SF) and surface pressure (SP).
- The same six variables from the ECMWF ERA5-Land reanalysis.
- Precipitation estimates from the National Oceanic and Atmospheric Administration (NOAA) Climate Prediction Center (CPC) Global Unified Gauge-Based Analysis of Daily Precipitation.
- Precipitation estimates from the NASA Integrated Multi-satellite Retrievals for GPM (IMERG) early run.
- Geological, geophysical and anthropogenic basin attributes from the HydroATLAS database[28].

All input data were area-weighted averaged over basin polygons over the total upstream area of each gauge or prediction point. The total upstream area for the 5,680 evaluation gauges used in this study ranged from 2.1 km$^2$ to 4,690,998 km$^2$.

No streamflow data were used as inputs to the AI model because (1) real-time data are not available everywhere, especially in ungauged locations, and (2) because the benchmark (GloFAS) does not use autoregressive inputs. We previously discussed how to use near-real-time target data in an AI-based streamflow model[31].

Extended Data Fig. 5 shows the time periods of available data from each source. During training, missing data was imputed either by using a similar variable from another data source (for example, HRES data were imputed with ERA5-Land data), or by imputing with a mean value and then adding a binary flag to indicate an imputed value, as described in ref. 31.

### Target and evaluation data

Training and test targets came from the Global Runoff Data Center (GRDC)[37]. Extended Data Fig. 6 shows the location of all streamflow gauges used in this study for both training and testing. We removed watersheds from the full, public GRDC dataset where drainage area reported by GRDC differed by more than 20% from drainage area calculated using watershed polygons from the HydroBASINS repository—this was necessary to ensure that poor-quality data, owing to imperfect catchment delineation, was not used for training. This left us with 5,680 gauges. Since we conducted the experiments reported in this paper, the GRDC has released catchment polygons for their gauge locations, so matching gauges with HydroBASINS watershed boundaries is no longer necessary.

### Experiments

We assessed the performance of the AI model using a set of cross-validation experiments. Data from 5,680 gauges were split in two ways. First, the data were split in time using cross-validation folds designed such that no training data from any gauge was used from within 1 year (the sequence length of the LSTM encoder) of any test data from any gauge. Second, the data were split in space using randomized (without replacement) $k$-fold cross-validation with $k = 10$. This pair of cross-validation processes were repeated so that all data (1984–2021) from all gauges were predicted in a way that was out-of-sample in both time and space. This avoids any potential for data leakage between training and testing. These cross-validation experiments are what is reported in the main text of this paper.

Other cross-validation experiments that we performed include splitting the gauge data in time, as above, and in space non-randomly according to the following protocol.
- Cross-validation splits across continents ($k = 6$).
- Cross-validation splits across climate zones ($k = 13$).
- Cross-validation splits across groups of hydrologically separated watersheds ($k = 8$), meaning that no terminal watershed contributed any gauges simultaneously to both training and testing in any cross-validation split.

The gauges in these cross-validation splits are shown in Extended Data Fig. 7. The results from these cross-validation splits are reported in Extended Data Figs. 8 and 9.

### GloFAS

GloFAS inputs are similar to the input data used in the AI model, with the main differences as follows.
- GloFAS uses ERA5 as forcing data, and not ERA5-Land.
- GloFAS (in the dataset used here) does not use ECMWF IFS as input to the model. (IFS data are used by the AI model for forecasting only, and we always compare with GloFAS nowcasts.)
- GloFAS does not use NOAA CPC or NASA IMERG data as direct inputs to the model.

GloFAS provides its predictions on a 3-arcmin grid (approximately 5-km horizontal resolution). To avoid large discrepancies between the

drainage area provided by the GRDC and the GloFAS drainage network, all GRDC stations with a drainage area smaller than 500 km² were discarded. The remaining gauges were geolocated on the GloFAS grid and the difference between the drainage area provided by the GRDC and the GloFAS drainage network was checked. If the difference between the drainage area was larger than 10% even after a manual correction of the station location on the GloFAS grid the station was discarded. A total of 4,090 GRDC stations were geolocated on the GloFAS grid.

In addition, unlike the AI model, GloFAS was not tested completely out-of-sample. GloFAS predictions came from a combination of gauged and ungauged catchments, and a combination of calibration and validation time periods. Extended Data Fig. 6 shows the locations of gauges where GloFAS was calibrated. This is necessary because of the computational expense associated with calibrating GloFAS, for example, over cross-validation splits. More information about GloFAS calibration can be found on the GloFAS Wiki[38].

This means that the comparison with the AI model favours GloFAS. Extended Data Fig. 9 shows scores using a set of standard hydrograph metrics in locations where GloFAS is calibrated, and can be compared with Extended Data Fig. 8, which shows the same metrics in all evaluation locations.

Although CEMS releases a full historical reanalysis (without lead times) for GloFAS version 4, long-term archive of reforecasts (forecasts of the past) of GloFAS version 4 do not span the full year at the time of the analysis. Given that reliability metrics must consider the timing of event peaks, this means that it is only possible to benchmark GloFAS at a 0-day lead time.

### Metrics

The results in the main text report precision and recall metrics calculated over predictions of events with magnitudes defined by return periods. Precision and recall metrics were calculated separately per gauge for both models. Return periods were calculated separately for each of the 5,680 gauges on both modelled and observed time series (return periods were calculated for observed time series and for modelled time series separately) using the methodology described by the US Geological Survey Bulletin 17b[39]. We considered a model to have correctly predicted an event with a given return period if the modelled hydrograph and the observed hydrograph both crossed their respective return period threshold flow values within two days of each other. Precision, recall and F1 scores were calculated in the standard way separately for each gauge. We emphasize that all models were compared against actual streamflow observations, and it is not the case that, for example, metrics were calculated directly by comparing hydrographs from the AI model with hydrographs from GloFAS. It is noted that it is possible for either precision or recall to be undefined for a given model at a given gauge owing to there being either no predicted or no observed events of a given magnitude (return period), and it is not always the case that precision is undefined when recall is undefined, and vice versa. This causes, for example, differences in the precision and recall sample sizes shown in Fig. 2.

All statistical significance values reported in this paper were assessed using two-sided Wilcoxon (paired) signed-rank tests. Effect sizes are reported as Cohen's term $d$[40], which is reported using the convention that the AI model having better mean predictions results in a positive effect size, and vice versa. All box plots show distribution quartiles (that is, the centre bar shows medians, not means) with error bars that span the full range of data excluding outliers. Not all results reported in this paper use all 5,680 gauges owing to the fact that some gauges do not have enough samples to calculate precision and recall scores over certain return period events. The sample size is noted for each result.

There are a large number of metrics that hydrologists use to assess hydrograph simulations[41], and extreme events in particular[42]. Several of these standard metrics are described in Extended Data Table 1 and

are reported for the models described in this paper in Extended Data Fig. 8, including bias, Nash–Sutcliffe efficiency (NSE)[43], and Kling–Gupta efficiency (KGE)[44]. KGE is the metric that GloFAS is calibrated to. Extended Data Fig. 9 shows the same metrics, but calculated over only gauges where GloFAS was calibrated (the AI model is still out-of-sample in these gauges). The results in Extended Data Figs. 8 and 9 show that the ungauged AI model is about as good in ungauged basins as GloFAS is in gauged basins when evaluated against the metrics that GloFAS is calibrated on (KGE), and is better in ungauged basins than GloFAS is in gauged basins on the (closely related) NSE metrics. However, GloFAS has better overall variance (the Alpha-NSE metric) than the ungauged AI model in locations where it is calibrated (although not in uncalibrated locations), indicating a potential way that the AI model might be improved.

### Data availability

Reanalysis (1984–2021) and reforecast (2014–2021) data produced by the AI model for this study, as well as corresponding GloFAS benchmark data, are available at https://doi.org/10.5281/zenodo.10397664 (ref. 45). Daily river discharge simulations are available for both GloFAS version 3 and GloFAS version 4 from the Climate Data Store[33]. For a summary of GloFAS versioning, see https://confluence.ecmwf.int/display/CEMS/GloFAS+versioning+system.

### Code availability

Fully functional trained models can be found at https://doi.org/10.5281/zenodo.10397664 (ref. 45). These trained models are runnable, but we lack the distribution license for the input data products, so to run them you must obtain and pre-process the relevant input data yourself. Input data can be obtained from the following sources: NASA IMERG precipitation data, https://gpm.nasa.gov/data; ECMWF HRES forecast data, https://www.ecmwf.int/en/forecasts/datasets/set-i; ECMWF ERA5-Land data, https://cds.climate.copernicus.eu/cdsapp#!/dataset/reanalysis-era5-land?tab=overview; NOAA CPC Global Unified Gauge-Based Analysis of Daily Precipitation data, https://psl.noaa.gov/data/gridded/data.cpc.globalprecip.html. In addition, the forecasting model developed for this project (along with several other AI streamflow forecasting models) was integrated into the NeuralHydrology code base[22] available at https://neuralhydrology.github.io. Using these research-grade models within the NeuralHydrology framework makes it easier to run conceptually similar models with your own input datasets. The code for reproducing the figures and analyses reported in this paper is available at https://github.com/google-research-datasets/global_streamflow_model_paper. This repository calculates metrics for the AI model and GloFAS outputs, as reported in this paper, and requires the Zenodo dataset[45].

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

**Acknowledgements** We thank P. Salamon at the European Commission's Joint Research Centre for providing GloFAS version 4 data, and for his insight with the analysis of that data.

**Author contributions** G.N. conducted experiments and analyses and wrote the first paper draft that was edited by all co-authors. G.S., F.K. and O.G. contributed substantially to experimental design and the design of the figures. All Google-affiliated authors contributed to development of the AI model. Authors with ECMWF affiliation (S.H., F.P. and C.P.) additionally helped to ensure proper processing of GloFAS data. S.N. completed the work while at Google. Y.M. supervised the research.

**Competing interests** The authors declare no competing interests.

**Additional information**
**Correspondence and requests for materials** should be addressed to Grey Nearing.

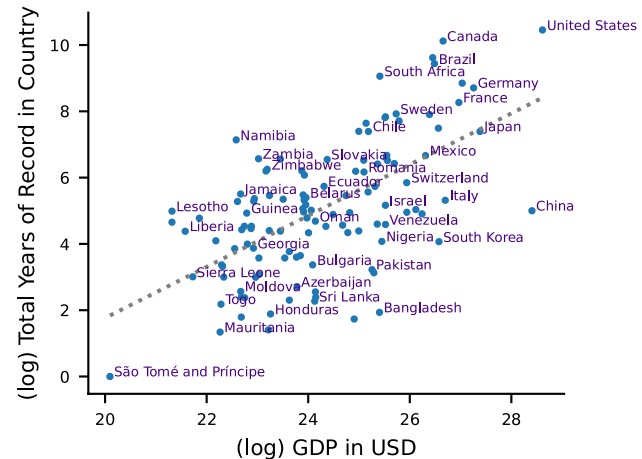

**Extended Data Fig. 1 | Streamflow data availability correlates with national GDP.** There is a log-log correlation (r=0.611; N=117) between national Gross Domestic Product (GDP) and the total number of years worth of daily streamflow data available in a country from the Global Runoff Data Center. GDP data are sourced from The World Bank[46].

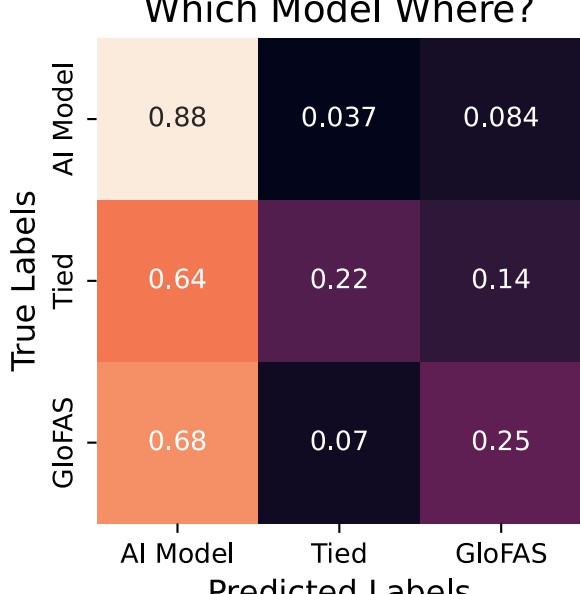

**Extended Data Fig. 2 | Confusion matrix of a classifier that predicts whether the AI model or GloFAS had a higher (or similar) F1 score in a given watershed based on geophysical catchment attributes (N = 3,360).** We found that this task is generally not possible given available catchment attribute data. Numbers shown on the confusion matrix are micro-averaged precision and recall, and colors serve as a visual indication of these same numbers. GloFAS simulation data from the Climate Data Store[33].

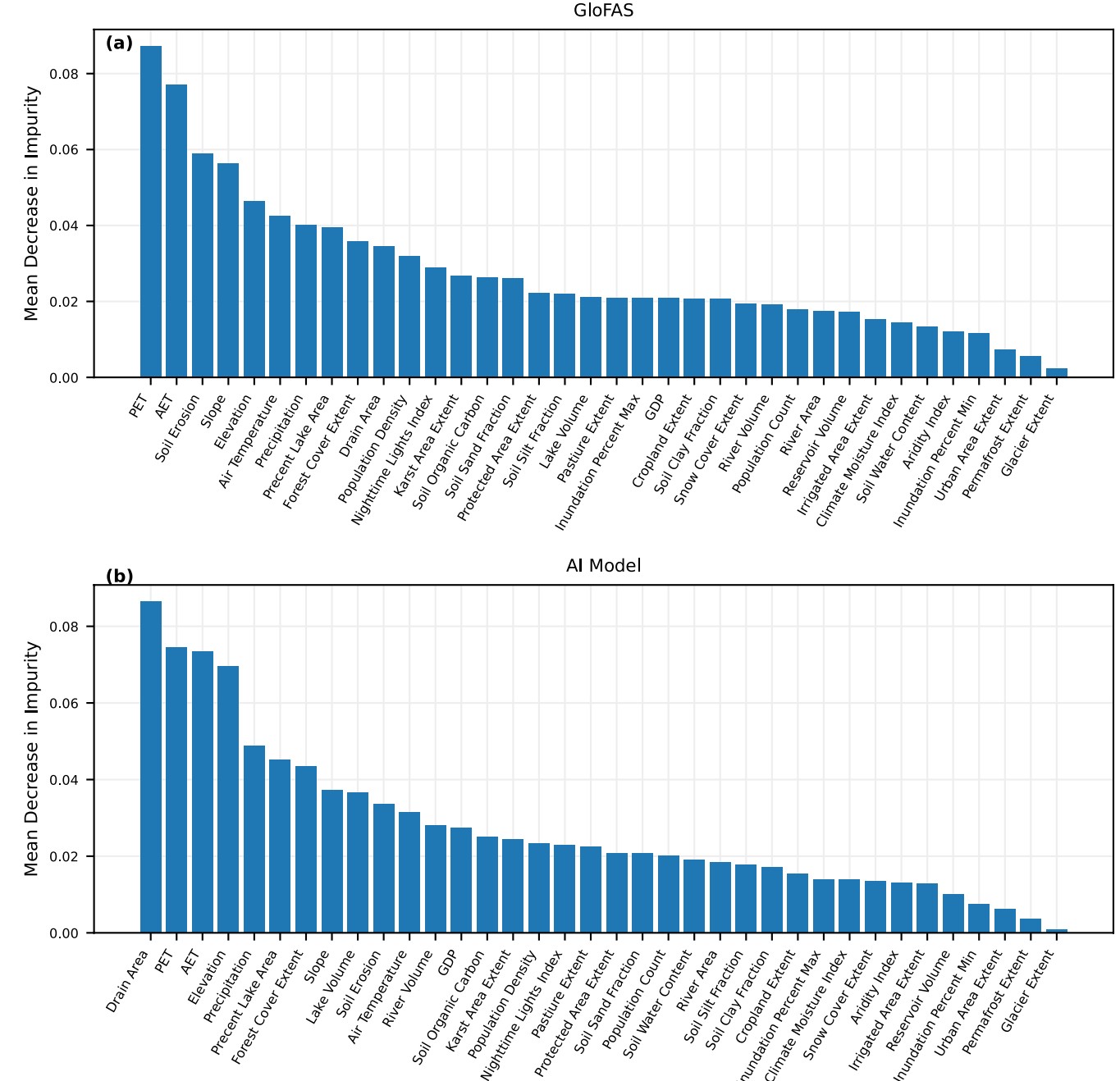

**Extended Data Fig. 3 | Full feature importance rankings of the score classifiers from Section 4 in the main paper.** These classifiers predict whether the GloFAS (panel a) or the AI model (panel b) performs better or worse than average in any given gauge location. The feature importance rankings shown here illustrate which catchment attributes the classifier uses to make those predictions. GloFAS simulation data from the Climate Data Store[33].

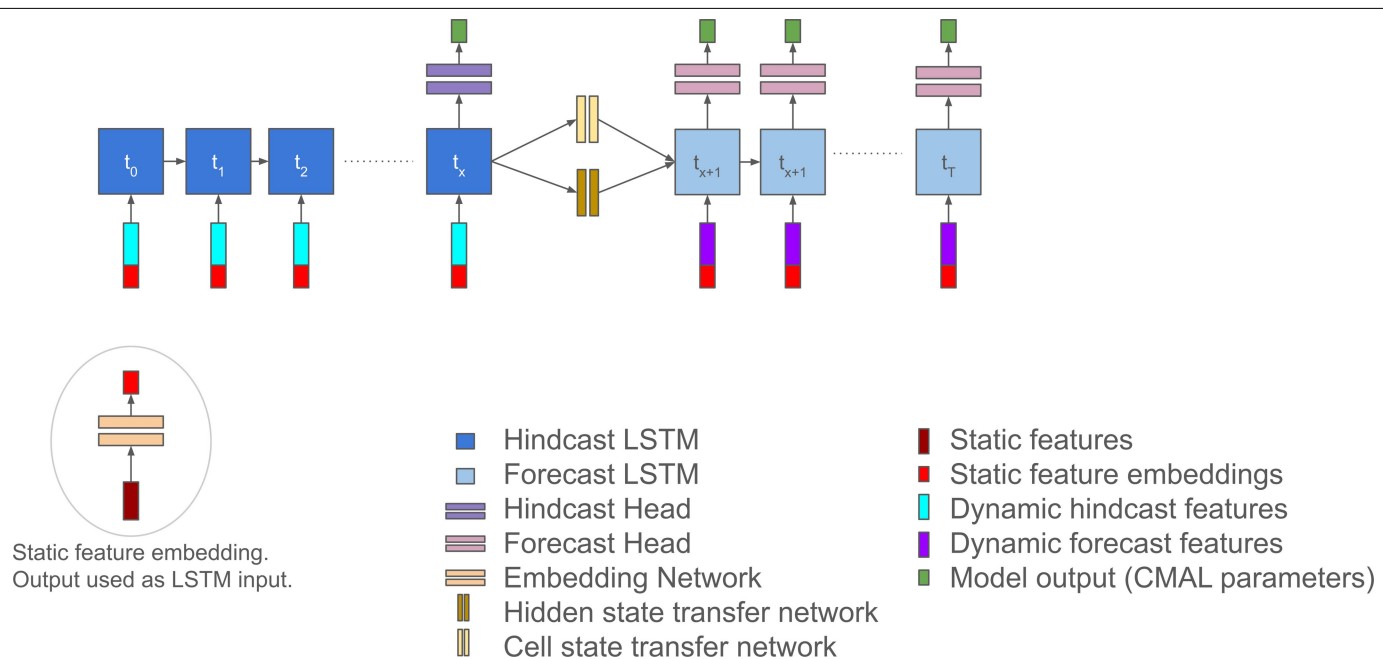

Static feature embedding.
Output used as LSTM input.

- ■ Hindcast LSTM
- ■ Forecast LSTM
- ▬ Hindcast Head
- ▬ Forecast Head
- ▬ Embedding Network
- ▯ Hidden state transfer network
- ▯ Cell state transfer network

- ▮ Static features
- ■ Static feature embeddings
- ▮ Dynamic hindcast features
- ▮ Dynamic forecast features
- ■ Model output (CMAL parameters)

**Extended Data Fig. 4 | Architecture of the LSTM-based forecast model developed for this project.** This is the model used operationally to support the Google Flood Hub https://g.co/floodhub.

**Extended Data Fig. 5 | Model input and training data.** Timeline showing the availability of each data source used for training and prediction with the AI model.

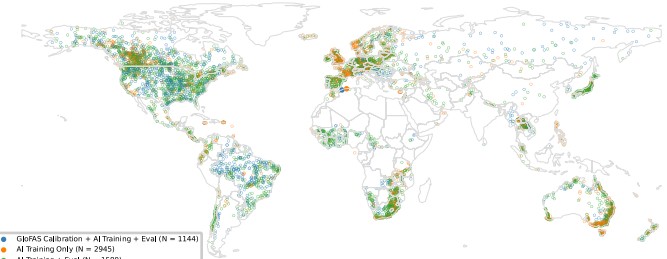

Training (Calibration) and Evaluation Gauge Locations

**Extended Data Fig. 6 | Location of gauges used for (i) training the AI model (*N* = 5,860), (ii) calibrating GloFAS (*N* = 1,144), and (iii) calculating the evaluation metrics reported in this paper (*N* = 4,089).** The AI model is a single model trained on data from all gauges simultaneously, while GloFAS was calibrated separately per-location and following a top-down approach from head-catchments to downstream catchments. All AI model evaluation was done out-of-sample in both location and time. Some of the 5,860 training gauges were excluded from evaluation because it was not possible to match those gauges to a GloFAS pixel. Basemap from GeoPandas[34].

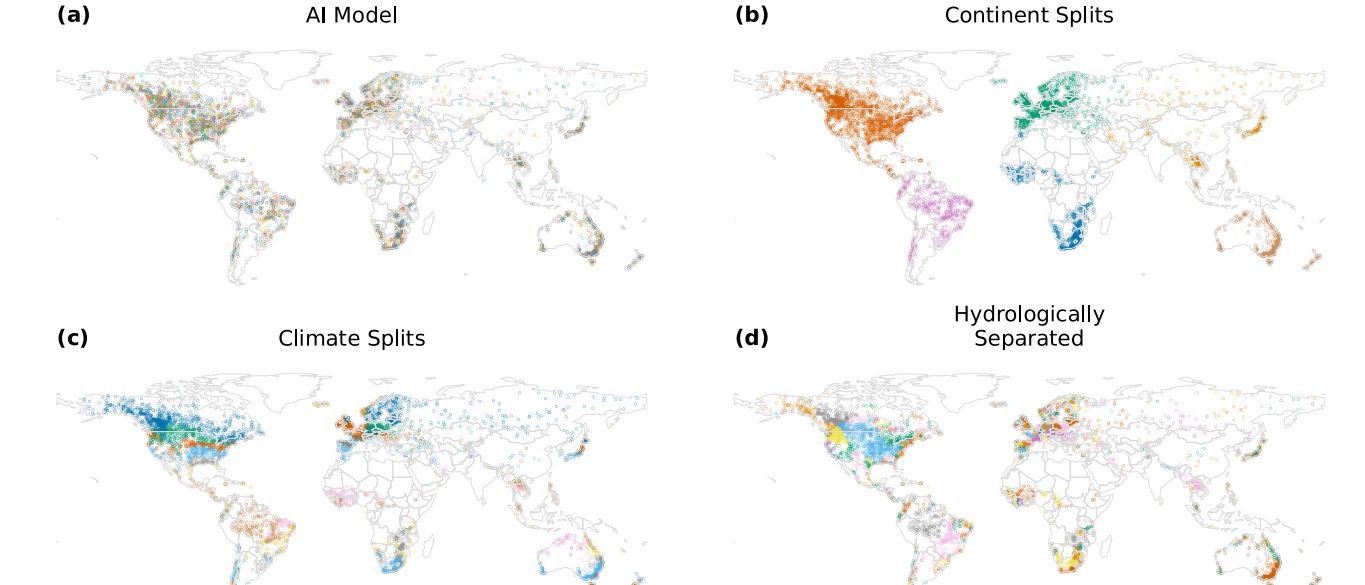

**(a)** AI Model

**(b)** Continent Splits

**(c)** Climate Splits

**(d)** Hydrologically Separated

**Extended Data Fig. 7 | Locations of gauges in each cross-validation split.** Different colors in each map represent different cross validation splits. Panel (a) shows random splits, which are the results reported in the main text of the paper. Panel (b) shows continent splits, so that all basins in a particular continent are in one cross validation group. Panel (c) shows climate zone splits, so that all basins in each of 13 climate zones are in one cross validation group. Panel (d) shows splits that group gauges in hydrologically-separated terminal basins. Basemaps from GeoPandas[34].

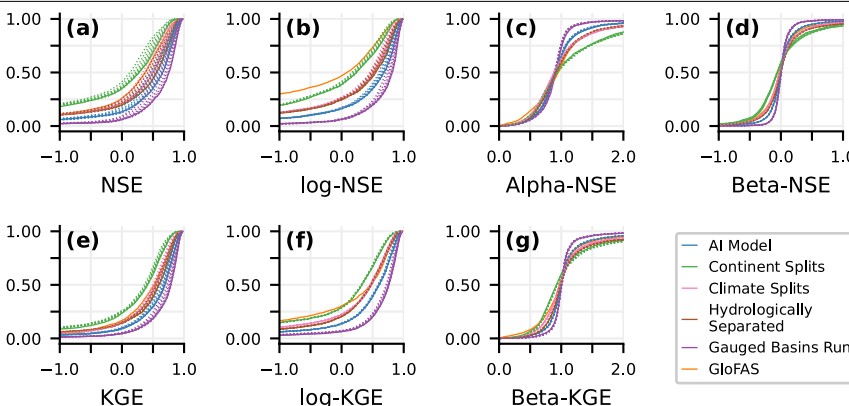

**Extended Data Fig. 8 | Hydrograph metrics for the AI model and GloFAS over all 4,089 evaluation gauges.** Cross validation splits are indicated by colors, and 0 to 7 day lead times are indicated by dashed lines (scores decrease with increasing lead time). Metrics are calculated on the time period 2014-2021. Metrics in panels (a-g) are listed in Extended Data Table 1. GloFAS simulation data from the Climate Data Store[33].

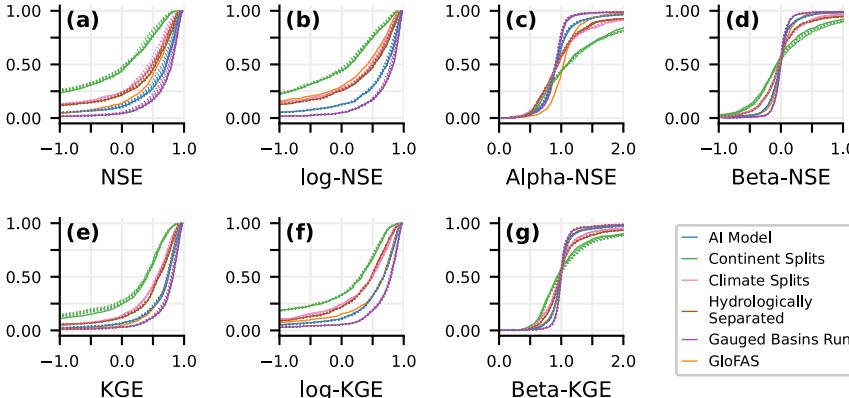

**Extended Data Fig. 9 | Hydrograph metrics for the AI model and GloFAS over the 1,144 gauges where GloFAS is calibrated.** Cross validation splits are indicated by colors, and 0 to 7 day lead times are indicated by dashed lines (scores decrease with increasing lead time). Metrics are calculated on the time period 2014-2021. Metrics in panels (a-g) are listed in Extended Data Table 1.

GloFAS is calibrated using the Kling-Gupta Efficiency (KGE), and when evaluated using this metric (as well as bias metrics), shows performance in *gauged* basins that is similar to the AI model in *ungauged* basins. GloFAS simulation data from the Climate Data Store[33].

**Extended Data Table 1 | A selection of standard hydrograph evaluation metrics**

| Metric | Description | Reference |
|---|---|---|
| NSE | Nash–Sutcliffe efficiency | Eq. 3 in[43] |
| log-NSE | Nash–Sutcliffe efficiency in logarithmic space | ~ |
| Alpha-NSE | Ratio of standard deviations of observed and simulated flow | Eq. 4 in[44] |
| Beta-NSE | Bias scaled by standard deviation of observations | Eq. 4 in[44] |
| KGE | Kling–Gupta efficiency | Eq. 9 in[44] |
| log-KGE | Kling–Gupta efficiency in logarithmic space | ~ |
| Beta-KGE | Ratio of mean simulated and mean observed flow | Eq. 10 in[44] |