## [Peer Review File · Nature]

Manuscript Title: Global prediction of extreme floods in ungauged watersheds

Reviewer Comments & Author Rebuttals

Reviewer Reports on the Initial Version:

Referees' comments:

Referee #1 (Remarks to the Author):

Review:

A. Summary of the key results

The authors presented an Artificial Intelligence (AI) model designed to predict extreme hydrological events up to 7 days in advance and compared its performance with the GloFAS model. The study found that the AI model significantly outperforms GloFAS across all continents, lead times, and return periods, especially in ungauged basins.

B. Originality and significance

The research emphasizes the efficacy of a global AI model in streamflow predictions, particularly at ungauged sites. While the comparison with GloFAS is noteworthy, the paper could benefit from examining how region-specific AI or physical models with direct streamflow data input might fare in contrast.

C. Data & methodology

The methodology, while innovative, presents a limitation. Although the exclusion of past streamflow data as inputs might cater to ungauged basins, it might hinder performance in locations where such data is available. The significance of spatial and temporal persistence in improving the accuracy of predictions could be underscored.

D. Appropriate use of statistics and treatment of uncertainties

The study highlights the relative performance against GloFAS, but a direct comparison to actual streamflow data would provide clearer insights into accuracy. Focusing predominantly on up to 10-year flood events might give a skewed perspective, especially as these events are typically more predictable in larger basins. It would be beneficial to elucidate the inputs used for both GloFAS and the AI model to understand any disparities or limitations in the input data.

E. Conclusions

The paper makes a strong case for the benefits of a global AI model, especially when pitted against the GloFAS model. Yet, the overarching conclusion might benefit from considering the potential of region-specific models, which might be better tailored to local conditions.

F. Suggested Improvements

Figure S2 lacks a legend, making it challenging to discern the significance of different colors on the map. Including this would enhance clarity and comprehension.

G. References

The authors have appropriately acknowledged previous works, drawing clear connections between their research and existing literature.

H. Clarity and context

The abstract and conclusion are both lucid and provide adequate context. The paper efficiently

conveys the significance of the research, its objectives, and the results achieved.

Overall, this paper makes valuable contributions to the understanding of flood predictions using AI models. With some clarifications and deeper examinations, particularly regarding data inputs and regional models, its impact could be even more pronounced.

Referee #2 (Remarks to the Author):

The study focuses on one of the hottest directions in flood forecasting, and its key contribution is the development of an artificial intelligence (AI) model that predicts extreme hydrological events with a lead time of 7 days. The model has been trained and validated globally.

1. The size and spatiotemporal accuracy of the data should be explicitly explained, whether it is in MB, GB, or Tb, as it is a crucial factor in determining the computation time for both training and validation periods. It is also important to provide information about the computational time metrics of the model.

2. While LSTM is indeed one of the representative intelligent models capable of capturing nonlinear features in time series data, the issue of its stability during runtime has always existed. It would be interesting to know how the authors address the robustness issue of AI models and whether there is any convergence of accuracy loss inherent to these models.

3. Detailed parameter settings for LSTM should be included in the experimental case, as hyperparameters such as the number of neurons, iterations, time steps, block size, etc., significantly impact the model's accuracy. It would be helpful to understand how the authors considered these parameters during calibration.

4. The claim that the developed AI model can be applied to predict extreme flood events should be supported by specific simulation results, including peak flow, flood volume, and flood hydrographs. These results can be added as supplementary material.

5. Existing research has shown that applying "max-min" normalization to input data can enhance the efficiency and accuracy of machine learning predictions. It would be interesting to know if the authors have considered this method to optimize the existing research findings.

6. It is well-known that hydraulic structures like reservoirs and dams significantly interfere with the consistency of river flow monitoring. Has the study fully considered and addressed these interference factors in the AI model prediction, or how were they handled?

7. There is text overlap in Figure S5 of the supplementary material. Please ensure proper formatting for better readability.

8. It is suggested to include a summary in Figure 9 that encompasses the overall information about input data spatiotemporal scales, methodology, processing steps, and output results. Currently, this holistic information is not apparent.

9. The paper mentions LSTM as the AI model, but it has been widely used in flood forecasting since 2018. I am curious if the authors have made any substantial improvements or modifications to the LSTM model in their research work.

10. AI models such as TCN, Transformer, and other parameter optimization algorithms also demonstrate excellent predictive capabilities. What is the rationale behind choosing LSTM as the core basis for this study's AI model? It would be more convincing to provide some comparative work in this regard.

11. Some figures need clear explanations of the axes' meanings. Please review and clarify them.

Referee #3 (Remarks to the Author):

This manuscript introduced an AI model to predict extreme hydrological events at timescales up to 7 days in advance, and compared the results with a traditional hydrology model named GloFAS.

As a researcher in the field of AI + hydrology, I am excited with a developing model globally

usable. This is novel and meaningful research and I have no concern about the input data, methodology, and results presented in this manuscript.

However, this manuscript primarily showcases statistical results and lacks in-depth analysis. At the same time, there is minimal discussion on AI-based forecasting. Authors discussed the ungauged basins in developing countries, but they seem not covered in this study due to a lack of data availability. In addition, this manuscript neither analyzes the possible reasons for poor performance in certain areas nor discusses directions for future plannings. I recommend major revisions.

Here are my comments and questions.

1. I have not used GloFAS before, but the official website introduced that GloFAS can provide global overview maps of flood probabilities up to 15 or 30 days in advance. May I ask if there is a reason you comparing AI-model only for lead time up to 7 days? If there is, they could be in Figure 4.

2. Can you provide more details about how you avoid data leakage on evaluating the return period events? I have a question on your cross validation apporhacn across 5,680 streamflow gauges. Are they randomly selected and distributed? Will the model performs different when there is upstream or downstream gauged? For example, if a stream has 3 guages, and only the gauge in the middle is selected as the validation gauge, it would be not surprise that it get a perfect performance using LSTM/Deep Learning.

3. The cross-validation studies conducted on various continents have piqued my interest. I'm eager to delve deeper into the analysis and ensuing discussion. Does this suggest that hydrological cycles on a global scale can be systematized, and that insights gained can be applied elsewhere?

4. You've presented an online platform that displays real-time data, including information on areas not discussed in the paper, such as India. However, your Figure 2 seems to focus only on regions where hydrological data is well-developed and has a long history of availability. I'm interested in understanding how this approach would apply to emerging economies with limited data history, as well as to truly ungauged basins.

5. The data in Figure 3 suggests that the AI model has shown substantial improvements compared to GloFAS. However, I did notice that both Precision and Recall reach 1.0 for the 2/5/10-year return periods. I'm curious if there's any analysis available on this data? This could potentially be provided in the supplementary material.

6. In Figure 5, you present four subgraphs for 1/2/5/10-year return periods. I distinctly observed that GloFAS scores on various continents tend to decrease in F1 score as the years increase. However, this is not consistently the case with the AI model. For instance, in the South West Pacific, the F1 score is below 0.4 for both 1 and 10-year return period events, but exceeds 0.5 for 2 and 5-year return period events. This pattern isn't very clear, and a similar situation is also observed in North America. Is there any discussion on this?

7. The analysis associated with Figure 6 seems rather cursory, so it's unsurprising that there aren't any compelling results. Floods can arise from a variety of causes—such as Coastal Floods, Snow-melt Floods, and Riverine/Flash floods, among others—each with its own unique factors. Predictions and analyses should accordingly take these differences into account. However, I didn't see any detailed hydrological analysis in this paper that would address these complexities.

8. The discussion surrounding Figure 7 (right) appears to be lacking. Does this imply that your model performs better in areas that are more urbanized (more light nigh index and flat areas) — perhaps due to a lack of snow-capped mountains—and conversely, less effectively in less urbanized regions?

9. Minor: There is Tied in Fig.6 but only above/below average in Fig.7 (left). I suggest to make them consistent.

10. Minor: I noticed an inconsistency in how the model is named across different figures. In most, it's referred to as the "AI model," but in Figure 7 (right), it's called "Google," and in Figure 8, it's termed "Google-ML." I assume these different names are referring to the same model, correct?

Response: All of the metrics presented in this paper, in both the main text and supplementary material, involve a direct comparison to actual streamflow observation data. We don't compare the AI model directly with GloFAS, we compare both models with observations. We added a sentence stating this explicitly in the first paragraph of the Metrics subsection in Methods.

Focusing predominantly on up to 10-year flood events might give a skewed perspective, especially as these events are typically more predictable in larger basins.

Response: The paper also includes standard hydrograph metrics (in supplementary material) that consider the entire hydrograph and do not depend on any specific return period. Our primary focus is on 2 and 5 year return period events, because these are commonly used as proxies for impactful flooding events.

It would be beneficial to elucidate the inputs used for both GloFAS and the AI model to understand any disparities or limitations in the input data.

Response: The inputs for the AI model are explained in Methods. In the revision, we added a description of the inputs to GloFAS.

E. Conclusions

The paper makes a strong case for the benefits of a global AI model, especially when pitted against the GloFAS model. Yet, the overarching conclusion might benefit from considering the potential of region-specific models, which might be better tailored to local conditions.

Response: If the reviewer is referring to regional models that are classical hydrology models (e.g., run by local agencies), the challenge is that we have not found long-term archives to work with. We alluded to this in the paper in the 5th paragraph of the introduction, however we avoided calling out anyone specifically for not archiving forecasts. Also, it is possible that there are some interesting forecast archives at some agency that we are not aware of. As an example, NOAA in the US does not have archives of past forecasts from their operational centers (the NWS RFCs). We previously used the NOAA National Water Model reanalysis as a benchmark, but that comparison is not interesting because the model is so bad (NOAA has started to replace the NWM).

If the reviewer is talking about regional ML models, we have looked at this extensively, and found that training a single global model is better. We expect to have a stand-alone publication documenting this research available on a preprint server (*Hydrology and Earth System Science Discussions*) likely before the reviewer reads this comment.

F. Suggested Improvements

Figure S2 lacks a legend, making it challenging to discern the significance of different colors on the map. Including this would enhance clarity and comprehension.

Response: Thank you. We added an explanation to the figure caption, which should be sufficient to understand the plot.

G. References

The authors have appropriately acknowledged previous works, drawing clear connections between their research and existing literature.

H. Clarity and context

The abstract and conclusion are both lucid and provide adequate context. The paper efficiently conveys the significance of the research, its objectives, and the results achieved.

Overall, this paper makes valuable contributions to the understanding of flood predictions using AI models. With some clarifications and deeper examinations, particularly regarding data inputs and regional models, its impact could be even more pronounced.

Referee #2 (Remarks to the Author):

The study focuses on one of the hottest directions in flood forecasting, and its key contribution is the development of an artificial intelligence (AI) model that predicts extreme hydrological events with a lead time of 7 days. The model has been trained and validated globally.

1、 The size and spatiotemporal accuracy of the data should be explicitly explained, whether it is in MB, GB, or Tb, as it is a crucial factor in determining the computation time for both training and validation periods. It is also important to provide information about the computational time metrics of the model.

Response: Thank you, this information was added to the Methods section of the revised paper.

2、 While LSTM is indeed one of the representative intelligent models capable of capturing nonlinear features in time series data, the issue of its stability during runtime has always existed. It would be interesting to know how the authors address the robustness issue of AI models and whether there is any convergence of accuracy loss inherent to these models.

Response: We are slightly unclear what the reviewer is referring to with regard to “stability during runtime” and “convergence of accuracy”. Assuming what the reviewer might mean (with apologies for any potential misunderstanding), our operational (and research) procedure is to train ensembles of LSTM models and combine the output. In our experience, there is no significant variability between the aggregate test/validation metrics of different training runs, indicating a convergence of accuracy. However, sometimes models do better or worse in different basins, which is why we use an ensemble. This ensemble averaging procedure is described in the Methods section.

3、 Detailed parameter settings for LSTM should be included in the experimental case, as hyperparameters such as the number of neurons, iterations, time steps, block size, etc., significantly impact the model's accuracy. It would be helpful to understand how the authors considered these parameters during calibration.

Response: This information is provided in the Methods section under the subsection “AI Model”. All of the major hyperparameters are explained in that section.

4、 The claim that the developed AI model can be applied to predict extreme flood events should be supported by specific simulation results, including peak flow, flood volume, and flood hydrographs. These results can be added as supplementary material.

Response: There are very many additional statistics that could be looked at. Unfortunately, space is very limited in this type of publication. We are preparing several different types of reports (blogs and articles for peer-review) that look at different aspects of flood predictability, including for large-scale and small scale analyses of individual flood events.

5、 Existing research has shown that applying "max-min" normalization to input data can enhance the efficiency and accuracy of machine learning predictions. It would be interesting to know if the authors have considered this method to optimize the existing research findings.

Response: We use standardization (subtracting mean and scaling by standard deviation) rather than min-max normalization on the input and target data. We added a sentence to this effect in the Methods section.

In general, min-max normalization is recommended when the data are bounded and/or no outliers exist. This is normally not the case for the time series that we consider. As an illustrative example, consider a time series of temperature measurements. If we were to use min-max normalization and get a colder year in the test-data than we observed during the training, then we could ingest negative values into the LSTM, which would be a potential problem since the network would have not previously seen negative inputs. We therefore prefer standardization.

6、 It is well-known that hydraulic structures like reservoirs and dams significantly interfere with the consistency of river flow monitoring. Has the study fully considered and addressed these interference factors in the AI model prediction, or how were they handled?

Response: We do not consider dams and reservoirs in the current version of our model. This is an area of future research.

7、 There is text overlap in Figure S5 of the supplementary material. Please ensure proper formatting for better readability.

Response: Thank you, this figure was updated and moved to the Extended Data section.

8、 It is suggested to include a summary in Figure 9 that encompasses the overall information about input data spatiotemporal scales, methodology, processing steps, and output results. Currently, this holistic information is not apparent.

Response: A full description of the input data, including preprocessing, is given in the Methods section. We did not reproduce all information (e.g., spatiotemporal resolution) about each input data product because that information is publicly available.

9、 The paper mentions LSTM as the AI model, but it has been widely used in flood forecasting since 2018. I am curious if the authors have made any substantial improvements or modifications to the LSTM model in their research work.

Response: The series of papers that (we believe) the reviewer is referring to (going back to 2018) largely do not include forecast models. Here we used an encoder/decoder LSTM for forecasting. The model is described in the Methods section, with details given in Figure 9.

10、 AI models such as TCN, Transformer, and other parameter optimization algorithms also demonstrate excellent predictive capabilities. What is the rationale behind choosing LSTM as the core basis for this study's AI model? It would be more convincing to provide some comparative work in this regard.

Response: First, we need to say that a comparison between ML modeling strategies is out of scope of this paper. To put it simply, we chose to scale and operationalize an LSTM-based model because this is the type of model that works best for this task.

We previously worked with both time convolutions and transformers for streamflow modeling. We believe that the reason that LSTM-based models are better is because the LSTM is Markovian and fits the problem better than these other models. We generally get about 1% - 2% better performance with a well-trained LSTM vs. a well-trained transformer. In addition to the fact that LSTMs are conceptually well-matched with this prediction problem, transformers generalize best for problems with large data sets, extensive pre-training tasks, and low noise (e.g., currently transformers scale better in language models than in image models). At present, it does not seem that streamflow hydrology is such a domain, although we would very much like to see the global streamflow data set continue to grow.

11、 Some figures need clear explanations of the axes' meanings. Please review and clarify them.

Response: Thank you, we added axis labels and sample sizes to all figures (to which these labels are relevant).

Referee #3 (Remarks to the Author):

This manuscript introduced an AI model to predict extreme hydrological events at timescales up to 7 days in advance, and compared the results with a traditional hydrology model named GloFAS.

As a researcher in the field of AI + hydrology, I am excited with a developing model globally usable. This is novel and meaningful research and I have no concern about the input data, methodology, and results presented in this manuscript.

However, this manuscript primarily showcases statistical results and lacks in-depth analysis. At the same time, there is minimal discussion on AI-based forecasting. Authors discussed the ungauged basins in developing countries, but they seem not covered in this study due to a lack of data availability. In addition, this manuscript neither analyzes the possible reasons for poor performance in certain areas nor discusses directions for future plannings. I recommend major revisions.

Response: Some public data is available (and was used) from the developing world. This lack of data is the main reason why prediction in ungauged basins is critical, and the main point of this paper is to show that AI provides significant improvement to prediction in ungauged basins. The paper includes an analysis of where the model performs better or worse based on geographical and geophysical watershed attributes.

Here are my comments and questions.

1. I have not used GloFAS before, but the official website introduced that GloFAS can provide global overview maps of flood probabilities up to 15 or 30 days in advance. May I ask if there is a reason you comparing AI-model only for lead time up to 7 days? If there is, they could be in Figure 4.

Response: We have found that offering longer-term forecasts can be misleading, since the skill drops off quickly due to uncertainty in meteorological forecasts. Our philosophy is that we prefer to issue forecasts at shorter lead times. Evaluation of the first 7 days of forecast is indicative of overall model performance, while later is more equivalent to climatology.

2. Can you provide more details about how you avoid data leakage on evaluating the return period events? I have a question on your cross validation approach across 5,680 streamflow gauges. Are they randomly selected and distributed? Will the model perform differently when there is upstream or downstream gauged? For example, if a stream has 3 gauges, and only the gauge in the middle is selected as the validation gauge, it would be not surprise that it get a perfect performance using LSTM/Deep Learning.

Response: The cross validation splits are chosen randomly (although see Figure S2 in supplementary material). However, please notice that there is no chance for data leakage because the cross validation experiments include a time split as well as a space split. There is no chance for any data seen during training to be present in any form during inference – this also means that no meteorological or hydrological event (e.g., a single storm that affects many gauges) can be present in both training and testing. Additionally, because there is no routing model, the model is unaware of spatial or hydrological relationships between basins, and is not able to directly transfer information along a stream channel (e.g., from an in-sample to out-of-sample gauge location). There are similarities between the catchment attributes of gauges that are on the same river, but there is no possibility for data leakage.

3. The cross-validation studies conducted on various continents have piqued my interest. I'm eager to delve deeper into the analysis and ensuing discussion. Does this suggest that hydrological cycles on a global scale can be systematized, and that insights gained can be applied elsewhere?

Response: Yes, we believe so. Please see this paper² that we wrote about a similar question.

4. You've presented an online platform that displays real-time data, including information on areas not discussed in the paper, such as India. However, your Figure 2 seems to focus only on regions where hydrological data is well-developed and has a long history of availability. I'm interested in understanding how this approach would apply to emerging economies with limited data history, as well as to truly ungauged basins.

Response: There is no difference in how the model would be applied to “truly” ungauged basins vs. how it is applied in this study to basins where data was withheld from training (the latter being necessary to have data to validate against).

We are working with a few partner countries, including India, who give us data for training, and also real time data that we use in our operational model. We exclude those cases from this study because they are not representative of overall model performance (they are usually better). Also, these analyses were excluded because some of the data is proprietary and would not contribute to open science.

5. The data in Figure 3 suggests that the AI model has shown substantial improvements compared to GloFAS. However, I did notice that both Precision and Recall reach 1.0 for the 2/5/10-year return periods. I'm curious if there's any analysis available on this data? This could potentially be provided in the supplementary material.

Response: Yes, there are some locations where we hit every 2, 5, or 10 year event, so the precision will be one. Similarly for recall. If the reviewer is interested in looking at these cases in

²Nearing, Grey S., et al. "What role does hydrological science play in the age of machine learning?." *Water Resources Research* 57.3 (2021): e2020WR028091.

particular in more depth, all of the data (and analysis code) from this paper are publicly available in the links provided in the paper.

6. In Figure 5, you present four subgraphs for 1/2/5/10-year return periods. I distinctly observed that GloFAS scores on various continents tend to decrease in F1 score as the years increase. However, this is not consistently the case with the AI model. For instance, in the South West Pacific, the F1 score is below 0.4 for both 1 and 10-year return period events, but exceeds 0.5 for 2 and 5-year return period events. This pattern isn't very clear, and a similar situation is also observed in North America. Is there any discussion on this?

Response: This is a correct interpretation of the results, and we did not include any specific discussion on this, as space is (extremely) limited and this is not, in our opinion, one of the major results.

7. The analysis associated with Figure 6 seems rather cursory, so it's unsurprising that there aren't any compelling results. Floods can arise from a variety of causes—such as Coastal Floods, Snow-melt Floods, and Riverine/Flash floods, among others—each with its own unique factors. Predictions and analyses should accordingly take these differences into account. However, I didn't see any detailed hydrological analysis in this paper that would address these complexities.

Response: As indicated in the introduction, this paper is about riverine (fluvial) floods. We added an additional indicator of this to the introductory paragraph.

8. The discussion surrounding Figure 7 (right) appears to be lacking. Does this imply that your model performs better in areas that are more urbanized (more light high index and flat areas)—perhaps due to a lack of snow-capped mountains—and conversely, less effectively in less urbanized regions?

Response: Our opinion is that there is no reason to speculate. We pointed out in this section that some of these catchment indicator correlations points toward possible future work.

9. Minor: There is Tied in Fig.6 but only above/below average in Fig.7 (left). I suggest to make them consistent.

Response: It does not make sense to add an “equals mean” option to a classifier, since this would be a bucket with 0 samples.

10. Minor: I noticed an inconsistency in how the model is named across different figures. In most, it's referred to as the "AI model," but in Figure 7 (right), it's called "Google," and in Figure 8, it's termed "Google-ML." I assume these different names are referring to the same model, correct?

Response: Thank you for catching this. Yes, they are the same model. This labeling is fixed in the revision.

Reviewer Reports on the First Revision:

Referees' comments:

Referee #1 (Remarks to the Author):

The authors have successfully addressed all previously noted comments from reviewers. The revised version of the manuscript is, therefore, can be acceptable for publication.

Referee #2 (Remarks to the Author):

The revised paper has made significant improvements, especially in terms of originality and significance, data, methodology, and conclusions. I think it can meet the standards of publication.

Referee #3 (Remarks to the Author):

This manuscript introduces an AI model to extreme predict hydrological events at timescales upto 7 days in advance, and compared the results with a traditional hydrology model named GIOFAS.

As a researcher in the field of AI + hydrology, I am excited with a developing model globally usable. Although LSTM models in hydrology have been widely studied and compared at the national level between physical and AI models. This is novel and meaningful research that first shown the application of these models on a global scale.

The authors have made a commendable effort in responding to the issues I initially raised, especially concerning comments #1-#4. The reply, results and data are trustworthy and are in line with previously published research as I known. Specifically the AI model shows an initial advantage over traditional physical models in shorter lead times, a benefit that gradually reduces and becomes less effective than physical models with longer lead times. This trend is consistent with the expected behavior of the LSTM model. Nonetheless, my concerns regarding the points raised in comments #5-7 have not been fully addressed. The manuscript does not provide clear insights and analyzes into the specific conditions under which the model excels. While the grouped data for different continents is presented, a comprehensive analysis based on hydrological environments is missing . For example, the paper from Hunt et al. (2022) analyzed and found that a pair of rain-on-snow events was not predicted by the GloFAS model but LSTM predicted them. The missing on the analysis is notable, especially since the LSTM model is shown to achieve perfect precision and recall in various cases. I find no further concerns regarding the methodology, data, or results presented in this paper.

In my view, this manuscript appears more like a product report rather than a conventional research paper. This manuscript may enhance the application of AI in hydrology and Earth sciences, thereby fostering the advancement of these fields. However, the lack of analysis reduces the value of this article in hydrological research.

Hunt, K. M., Matthews, G. R., Pappenberger, F., & Prudhomme, C. (2022). Using a long short-term memory (LSTM) neural network to boost river streamflow forecasts over the western United States. *Hydrology and Earth System Sciences* , 26(21), 5449-5472.

Response to Reviewers

As a researcher in the field of AI + hydrology, I am excited with a developing model globally usable. Although LSTM models in hydrology have been widely studied and compared at the national level between physical and AI models. This is novel and meaningful research that first shown the application of these models on a global scale.

The authors have made a commendable effort in responding to the issues I initially raised, especially concerning comments #1-#4. The reply, results and data are trustworthy and are in line with previously published research as I know. Specifically the AI model shows an initial advantage over traditional physical models in shorter lead times, a benefit that gradually reduces and becomes less effective than physical models with longer lead times. This trend is consistent with the expected behavior of the LSTM model.

We thank the reviewer for their time and effort, and also for their kind words. One thing we would like to point out is that our results don't indicate that benefit gradually decreases with lead time. Our results show that the benefit *relative to GloFAS nowcasts* decreases with lead time, but that is a much higher bar than comparing with GloFAS at similar lead times. Unfortunately, GloFAS does not archive full historical re-forecasts, so it is not possible to benchmark using their data at lead times greater than zero.

Nonetheless, my concerns regarding the points raised in comments #5-7 have not been fully addressed. The manuscript does not provide clear insights and analyzes into the specific conditions under which the model excels. While the grouped data for different continents is presented, a comprehensive analysis based on hydrological environments is missing. For example, the paper from Hunt et al. (2022) analyzed and found that a pair of rain-on-snow events was not predicted by the GloFAS model but LSTM predicted them. The missing on the analysis is notable, especially since the LSTM model is shown to achieve perfect precision and recall in various cases. I find no further concerns regarding the methodology, data, or results presented in this paper.

While we sympathize with the reviewer about this, space is limited in an article like this. We devoted quite a large fraction of the available article space to analyzing where the model performs well vs poorly (this is one of three main sections in the main body of the text, with further analysis in Extended Data). The approach we used to do this is automated and aggregated, instead of anecdotal (the latter being a common approach in hydrology literature). This choice was necessary because analyzing anecdotally over several thousand gauges is neither feasible nor representative.

In my view, this manuscript appears more like a product report rather than a conventional research paper. This manuscript may enhance the application of AI in hydrology and Earth sciences, thereby fostering the advancement of these fields. However, the lack of analysis reduces the value of this article in hydrological research.

Hunt, K. M., Matthews, G. R., Pappenberger, F., & Prudhomme, C. (2022). Using a long short-term memory (LSTM) neural network to boost river streamflow forecasts over the western United States. *Hydrology and Earth System Sciences* , 26(21), 5449-5472.